# How Has Prostate Cancer Radiotherapy Changed in Italy between 2004 and 2011? An Analysis of the National Patterns-Of-Practice (POP) Database by the Uro-Oncology Study Group of the Italian Society of Radiotherapy and Clinical Oncology (AIRO)

**DOI:** 10.3390/cancers13112702

**Published:** 2021-05-30

**Authors:** Ercole Mazzeo, Luca Triggiani, Luca Frassinelli, Alessia Guarneri, Sara Bartoncini, Paolo Antognoni, Stefania Gottardo, Diana Greco, Simona Borghesi, Sara Nanni, Alessio Bruni, Gianluca Ingrosso, Rolando Maria D’Angelillo, Beatrice Detti, Giulio Francolini, Alessandro Magli, Andrea Emanuele Guerini, Stefano Arcangeli, Luigi Spiazzi, Umberto Ricardi, Frank Lohr, Stefano Maria Magrini

**Affiliations:** 1Radiotherapy Unit, Oncology and Hematology Department, University Hospital of Modena, 41124 Modena, Italy; mazzeo.ercole@aou.mo.it (E.M.); frassinelli.luca@aou.mo.it (L.F.); lohr.frank@aou.mo.it (F.L.); 2Radiation Oncology Department, University and Spedali Civili Hospital, 25123 Brescia, Italy; luca.triggiani@unibs.it (L.T.); diana.greco@unibs.it (D.G.); a.guerini012@unibs.it (A.E.G.); stefano.magrini@unibs.it (S.M.M.); 3Department of Oncology, Radiation Oncology, Azienda Ospedaliero-Universitaria Città della Salute e Della Scienza, 10126 Turin, Italy; aguarneri@cittadellasalute.to.it (A.G.); sbartoncini@cittadellasalute.to.it (S.B.); 4Radiotherapy Deparment, ASST dei Sette Laghi-Ospedale di Circolo e Fondazione Macchi, 21100 Varese, Italy; paolo.antognoni@asst-settelaghi.it; 5Service of Radiotherapy, Istituito Clinico Sant’Ambrogio, 25123 Milan, Italy; stefania.gottardo@grupposandonato.it; 6Radiation Oncology Unit of Arezzo-Valdarno, Azienda USL Toscana Sud Est, 52100 Arezzo, Italy; simona.borghesi@uslsudest.toscana.it (S.B.); sara.nanni@uslsudest.toscana.it (S.N.); 7Radiation Oncology Section, Department of Medicine and Surgery, University of Perugia, 06123 Perugia, Italy; gianluca.ingrosso@unipg.it; 8Depatment of Biomedicine and Prevention, Tor Vergata University of Rome, 00133 Rome, Italy; d.angelillo@med.uniroma2.it; 9Unit of Radiation Oncology, Azienda Ospedaliero-Universitaria Careggi, 50134 Florence, Italy; beatrice.detti@aouc.unifi.it (B.D.); francolinigiulio@gmail.com (G.F.); 10Department of Radiation Oncology, Udine General Hospital, 33100 Udine, Italy; maglialessandro@hotmail.com; 11Department of Radiation Oncology, S. Gerardo Hospital—University of Milan Bicocca, 20900 Monza, Italy; stefano.arcangeli@unimib.it; 12Department of Medical Physics, Spedali Civili Hospital, 25123 Brescia, Italy; luigi.spiazzi@unimi.it; 13Department of Oncology, Radiation Oncology, University of Turin, 10126 Turin, Italy; umberto.ricardi@unito.it

**Keywords:** Pattern Of Practice, prostate cancer, radiotherapy

## Abstract

**Simple Summary:**

This is a safety and efficacy analysis from a very large dataset of patients affected by localized prostate cancer having received radiotherapy with or without concomitant androgen deprivation therapy in twelve academic and non-academic Italian Institutions. The aim of this retrospective "real life" study was to provide additional data on clinical presentation, diagnostic workup, radiation therapy management and toxicity as collected within the framework of POP III. Though the usual limitations for a retrospective analysis apply, it nevertheless may expand the current knowledge in this area showing the progress of radiation therapy techniques and clinical outcomes in the period between 2004 and 2011 after a significant period of follow up.

**Abstract:**

Background and purpose: Two previous “Patterns Of Practice” surveys (POP I and POP II), including more than 4000 patients affected by prostate cancer treated with radical external beam radiotherapy (EBRT) between 1980 and 2003, established a “benchmark” Italian data source for prostate cancer radiotherapy. This report (POP III) updates the previous studies. Methods: Data on clinical management and outcome of 2525 prostate cancer patients treated by EBRT from 2004 to 2011 were collected and compared with POP II and, when feasible, also with POP I. This report provides data on clinical presentation, diagnostic workup, radiation therapy management, and toxicity as collected within the framework of POP III. Results: More than 50% of POP III patients were classified as low or intermediate risk using D’Amico risk categories as in POP II; 46% were classified as ISUP grade group 1. CT scan, bone scan, and endorectal ultrasound were less frequently prescribed. Dose-escalated radiotherapy (RT), intensity modulated radiotherapy (IMRT), image guided radiotherapy (IGRT), and hypofractionated RT were more frequently offered during the study period. Treatment was commonly well tolerated. Acute toxicity improved compared to the previous series; late toxicity was influenced by prescribed dose and treatment technique. Five-year overall survival, biochemical relapse free survival (BRFS), and disease specific survival were similar to those of the previous series (POP II). BRFS was better in intermediate- and high-risk patients treated with ≥ 76 Gy. Conclusions: This report highlights the improvements in radiotherapy planning and dose delivery among Italian Centers in the 2004–2011 period. Dose-escalated treatments resulted in better biochemical control with a reduction in acute toxicity and higher but acceptable late toxicity, as not yet comprehensively associated with IMRT/IGRT. CTV-PTV margins >8 mm were associated with increased toxicity, again suggesting that IGRT—allowing for tighter margins—would reduce toxicity for dose escalated RT. These conclusions confirm the data obtained from randomized controlled studies.

## 1. Introduction

Prostate Cancer (PCa) is the second most frequently diagnosed tumor in the male population worldwide [1]. In Italy, it represents approximately 19% of all tumors diagnosed in men aged 50 and older, excluding skin cancers [2].

Radiation therapy (RT) is a very effective treatment modality for PCa. The introduction of new treatment planning and dose delivery technology further strengthened its role as a radical and postoperative treatment, allowing the implementation of dose escalation protocols and hypofractionated radiotherapy (HypoRT) with acceptable rectal and genitourinary toxicity. The clinical impact of image-guided radiotherapy (IGRT) techniques has already been tested in randomized trials, highlighting an improved tumor control and decreased toxicity, resulting in a favorable therapeutic index [3,4]. In 2002 and 2013, two different Italian studies were published assessing the patterns of practice (POP) of PCa patients in Italy [5,6]. The current study was performed by the Uro-Oncology Study Group of the Italian Society of Radiotherapy and Clinical Oncology (AIRO) to assess the diffusion of new technologies such as IMRT/VMAT and IGRT and their impact on clinical outcome and toxicity in the 2004–2011 period.

## 2. Materials and Methods

In an iterative process, the Uro-Oncology Study Group of AIRO defined the shared database items for the 2004–2011 observation period, taking into account those previously investigated in the first two surveys. To simplify the description of the results, the previously published series have been labeled POP I (1995–1998 period) and POP II (1999–2003 period), while the current series is referred to as POP III. Five academic and seven non-academic hospital-based centers located in the central and northern Italian regions participated in the current study; among them, four participated in POPI and POPII, four participated only in POPII, and the remaining four only in POPIII. A web-based database was created to collect patient, tumor, diagnostic, treatment, and technical features.

Data on 2525 prostate cancer patients treated consecutively with radical EBRT in association or not with androgen deprivation therapy (ADT) from 2004 to 2011 were collected. All patients were treated according to the individual center protocols, all aligned with the available national and international guidelines. Patients were followed at each center for a period of at least 5 years in an out-patient clinic, every 3–6 months during the first 2–3 years and every 6–12 months thereafter. The retrospective data collection procedure was approved by the Ethical Committee of the coordinating center (Modena, practice n° 231/13) and subsequently by those of each participating center. Follow-up information was retrieved from patient clinical records (and follow-up visits, when needed to update charts in living patients). Acute and late GI and GU toxicity rates (scored by the RTOG toxicity scale) [7] were recorded at the different follow up intervals, and the most severe toxicity experienced by the patients was reported for the purpose of this analysis.

Because this is a retrospective data collection based on a large multi-institutional database, it was not possible to retrieve all data related to some of the analyzed variables. In the Results section, all missing data are, however, clearly identified. Not all the database items were present in all three series. To better elucidate the progressive diffusion of the technical innovations, the study period was also divided into two consecutive four-year periods; the number of patients treated was almost equally distributed between the 2004–2007 quadriennium (1246, 49%) and the subsequent one (2008–2011, 1279, 51%).

### Statistical Analysis

A descriptive analysis of POP III was performed and data were compared to those from POP I and POP II. To find significant differences for some categorical variables, the Chi-squared test was used, while for continuous (scale) variables the T-test and the one-way ANOVA were used. Cancer-specific death, death by any cause and biochemical failure by ASTRO criteria [8] (the same used in POPI and POPII), calculated from the end of RT, were analyzed with the Kaplan–Meier method (KM) to obtain five-year disease-specific (DSS), overall survival (OS), and biochemical relapse-free survival (BRFS) rates for the whole series. KM was also used to evaluate patients’ rate of late side effects, and the log-rank test was used to compare the risk of biochemical failure between D’Amico risk groups. The statistical analysis was performed by using SPSS vv 24©. Graphs were created with Graph Pad prism 6©.

## 3. Results

### 3.1. Patients’ Clinical Features 

Median age was 72 years (range, 46–88), median follow-up was 6 years (range 0–13). More than 50% (1495, 59%) of patients presented with low or intermediate D’Amico risk disease, in line with the POP II (52%) cohort, while 1016 patients (40%) presented with high risk disease. Mean PSA at diagnosis was 14.14 ng/mL ± 19.03 SD. Cases with PSA values > 20 ng/mL dropped from 32% (POP I) to 27% (POP II) to 15% (POP III).

It was possible to calculate the ISUP group for most POP II and POP III patients [9]. 1152 POP III patients (46%) were classified as Group 1, 548 (22%) as Group 2, 254 (10%) as Group 3, 356 (14%) as Group 4, and 188 (7%) as Group 5. While only relatively small differences were observed in the distribution of the ISUP group, a Gleason score between 2–5 was reported in only 6% of the POP III cohort, as opposed to 27% of the POP II cases (*p* < 0.0001); this observation parallels the 11% increase in the proportion of patients with Gleason sum ≥ 7 in the POP III series.

More than 60% of the POP III patients showed clinically evident disease being classified as cT2 or cT3 using the TNM [10]. Regarding nodal status, only 2% of the cases had positive nodes. Patient’ features are reported in Table 1.

### 3.2. Diagnostic Workup 

Significant differences were found between POP II and POP III regarding diagnostic staging procedures before treatment (Table 2). Abdominal/pelvic CT scans (73% vs. 53%) (*p* = 0.001) as well as transrectal ultrasound (87% vs. 68%) (*p* = 0.001) and bone scans (84% vs. 53%) (*p* = 0.001) were used in fewer patients in POP III than in POP II. No statistically significant difference was found for the use of magnetic resonance imaging. Only a small number of POP III patients (2%) was staged with choline PET; this procedure was not used at all among POP I and POP II patients.

### 3.3. Treatment Features

Prescription dose (Figure 1) in patients who underwent conventionally fractionated RT (1.8–2 Gray [Gy]/fraction) clearly increased when compared to the previous series, as 1136 patients of POP III were given a total dose ≥ 76 Gy (45% vs. 9% in POP II) (*p* = 0.001), and the mean dose to the prostate (74.1 Gy) was significantly higher (one-way ANOVA, *p* < 0.0001) than in POP II (71.6 Gy) and POP I (69 Gy). Less than 1% of patients treated with conventional fractionation received doses < 70 Gy (Table 3).

All patients were treated with conformal RT; 1957 patients (78%) were treated with three-dimensional conformal RT while 568 (22%) underwent IMRT.

IGRT also started to be available and was used in 337 patients (13%) recruited in the POP III study.

The use of HypoRT is reported for the first time in the current series; 20 patients (<1%) underwent a moderately HypoRT regimen (dose per fraction 2.2–2.3 Gy), 47 patients (2%) were treated with higher dose per fraction (2.5–3.1 Gy), but with a total dose < 70 Gy, and 40 patients (2%) underwent a shorter HypoRT schedule (dose per fraction 2.7 Gy) with a total dose higher than 70 Gy.

As widely accepted [11,12], we considered a prostate tumor α/β ratio of 1.5 to obtain normalized equivalent dose in 2-Gy fractions (EqD2) using the linear quadratic model. The mean equivalent dose (EqD2) to the prostate overall was 74.5 Gy (hypofractionated regimens included).

Mean dose (EqD2) to the prostate with HypoRT was 80.7 Gy, significantly higher (one-way ANOVA, *p* <0.0001) than the conventionally fractionated schedules.

HypoRT was predominantly delivered with IGRT (86% vs. 10% of conventional regimens) and IMRT (97% vs. 19% of conventional regimens) techniques.

Mean dose (EqD2) to the prostate in IMRT treated patients was 77 Gy, significantly higher than in patients treated with 3D-CRT (74 Gy, *t*-test *p* = 0.04). Patients treated with IGRT techniques received a mean dose (EqD2) to the prostate significantly higher (76 Gy) than those treated without IGRT (74 Gy, *t*-test *p* < 0.0001).

Almost all patients (2416, 95%) in the current series underwent treatment to the prostate and/or seminal vesicles, while prophylactic pelvic irradiation was delivered only to 107 patients (4%). Median overall treatment time was 54 days (SD ± 10). The photon beam energy mainly used was ≤10 MV (1479, 59%). More than 90% of the patients were treated in supine position with filled bladder and empty rectum. Isotropic CTV to PTV expansion was used in more than 50% of the patients. When non-isotropic expansion was performed, the posterior margin was reduced.

Most patients (70%) were treated with a CTV-PTV margin ≥ 8 mm. A smaller CTV-PTV margin was used when IGRT was given (70% vs. 19%, *p* < 0.0001); about 50% of the patients treated with the IMRT technique had CTV-PTV margins >8 mm (Figure 2).

A larger CTV-PTV margin (≥8 mm) was used more frequently among patients who received a total dose (EQD2) ≤ 74 Gy than in those receiving >74 Gy (82% vs. 64%, *p* < 0.0001).

By dividing the series into two consecutive time periods (2004–2007 and 2008–2011), it clearly emerges that the most modern techniques (IMRT and IGRT) were used more frequently during the second quadriennium and allowed an increase in the dose delivered as well as a reduction in the CTV-PTV margins (Table 4).

Concomitant androgen deprivation therapy (ADT) was administered to the vast majority of the POP III patients (76%), similar to what was previously reported for POP I and POP II. Data about the type of ADT was not available for 635 patients (25%). Of the remaining 1890, 578 (30.6%) had total androgen blockade, 592 (31.3%) were treated with LH-RH analogues, and 720 (38%) were treated with a peripheral antiandrogen alone. The mean duration of ADT was 18 months.

### 3.4. Acute Toxicity

Acute genito-urinary (GU) and gastrointestinal (GI) toxicity data were available for almost all patients (missing data less than 1%). The treatment was generally well tolerated because most patients experienced no or mild, clinically insignificant toxicity.

Grade 2 and 3 acute GU toxicity was reported in 343 patients (13%) and 30 patients (1.2%), respectively, while grade 4 GU acute toxicity was reported in only 3 patients.

No grade 4 acute GI toxicity was reported; G2 acute GI toxicity was observed in 260 patients (10%), while only 22 patients (<1%) experienced G3 toxicity.

Comparing the crude acute toxicity rate of the current series with that of POP II, there is a significantly (*p* < 0.0001) lower frequency of G2 cases in favor of the recent cohort, while data were similar for both G3-G4 GI and GU toxicity (Figure 3).

In the POPIII series, patients treated with IMRT experienced a lower incidence of GU toxicity (every grade, 55% vs. 67%, *p* < 0.001). Not considering the dose level, IGRT per se did not impact on the occurrence of both GU and GI acute toxicity (every grade); notably, however, no G4 GU toxicity was reported in IGRT treated patients as opposed to 3 G4 events in the non-IGRT group.

Patients treated with a CTV-PTV margin ≥ 8 mm experienced a higher incidence of acute GU toxicity (every grade, 66% vs. 60%, *p* < 0.01), whereas no significant impact on GI toxicity was demonstrated.

Dosimetric data were available for more than 50% of the patients. Among them the volume of rectum receiving 25% of the prescribed dose (V25^r^) and the dose at 50% of the rectal volume (D50^r^) were significantly greater in patients experiencing acute GI toxicity (every grade) (*t*-test, *p* < 0.001). Dose at 50% of the bladder volume (D50^b^) was significantly higher in patients experiencing acute GU toxicity (every grade, *t*-test, *p* < 0.001).

Counterintuitively, patients treated with a total dose (EQD2) ≤ 74 Gy experienced a higher incidence of GU (≥G2 grade) acute side effects than those who received >74 Gy (59% vs. 41% *p* < 0.001); a similar result was found for GI (every grade) toxicity (51% vs. 27%, *p* < 0.0001).

Comparing the crude rates of acute toxicity in the two consecutive four-year periods of the study, a clear relationship seems to emerge between an increase in treatments with IMRT, IGRT, CTV-PTV margins ≤ 8 mm in the more recent quadriennium and a significant reduction in acute GI toxicity incidence (Table 4). In addition, a non-significant reduction in acute GU toxicity was observed in the study years 2008–2011; of note, in this period, a significant increase in the fraction of patients treated with doses in excess of 78 Gy (EqD2) and in the mean EqD2 dose has been observed (Table 4).

### 3.5. Late Toxicity

Late toxicity data were available for nearly 70% (1708) of the patients.

Cumulative probability of five-year late toxicity (GU, GI, every grade) was 30%.

Cumulative probability of five-year late GU and GI toxicity ≥ G2 was 12% and 15%, respectively.

Late G2 and G3 GI toxicity was observed in 194 (7.7%) and 29 (1.1%) patients, respectively; only 1 patient had G4 toxicity.

Late G2, G3, and G4 GU toxicity was registered respectively in 164 (7.7%), 13 (0.5%), and 6 (0.2%) patients.

Not unexpectedly (because the average doses in POP III patients were higher than in POP II) the crude rates of G2 GI and G2 GU late toxicities were higher for POP III patients than in the previous series, while the percentage of patients who developed G3-G4 GI and GU toxicity was similar among the three series (Figure 4).

No significant impact of IMRT and IGRT (considered as dichotomous variables: yes/no) on both GI and GU late toxicity was observed.

Patients treated with a CTV-PTV margin ≥8 mm experienced a higher incidence of both late GI toxicity (every grade) (35% vs. 26%, chi-squared *p* < 0.0001), and late GU (≥G2) (12% vs. 9%, chi-squared *p* = 0.02).

A dosimetric evaluation on available data was also performed for late toxicity; V25^r^_,_ D50^r^ were significantly higher in patients who experienced late GI toxicity (every grade; *t*-test, *p* < 0.001). None of the available dosimetric data was instead related to an increase in GU late toxicity.

As described for acute toxicity, patients treated with a total dose (EQD2) ≤ 74 Gy experienced a higher incidence of GU late toxicity (every grade) than those who received >74 Gy (39% vs. 24%, chi-squared *p* < 0.001); a similar result was found for GI late toxicity (every grade) (38% vs. 19%, chi-squared *p* < 0.0001).

As for acute toxicity, we compared the crude rates of late GU and GI toxicity (≥G2) between the 2004–2007 and 2008–2011 study periods; no significant difference was found for GI toxicity, while there was a significant decrease of GU toxicity in the more recent period (Table 4). As already noted, in this period, a significant increase in the fraction of patients treated with IGRT, IMRT, CTV-PTV margins ≤ 8 mm, doses in excess of 78 Gy (EqD2), and in the mean EqD2 dose has been observed.

### 3.6. Overall Survival (OS) and Biochemical Relapse-Free Survival (BRFS)

Five year OS for the entire current series was 88%, equal to that reported for POP II (88%) and higher than that reported for POP I (77%). Five-year DSS was 97% as opposed to 96% for POP II and 86% for POP I. Five-year BRFS (ASTRO definition) was 86%, in line with POP II data (88%); data were not available for POP I. As expected, considering D’Amico risk classification, five-year BRFS was 96%, 88%, and 78% for low, intermediate, and high risk patients, respectively (log-rank, *p* < 0.0001). Intermediate and high risk patients (D’Amico classification) treated with a total dose (EqD2) ≥ 76Gy enjoyed an increased five-year BRFS (Figure 5) (95% vs. 86% for intermediate risk patients—84% vs. 76% for high risk patients, log-rank, *p* < 0.0001)

## 4. Discussion

Surgery and RT can both be considered standard treatments for localized PCa [13,14,15,16]. In Italy, however, surgery still remains the most frequently used therapeutic approach for localized malignancies, especially for younger patients and those without relevant comorbidities [17]. From a radiation oncologists’ perspective, recent technological innovations and the results of dose escalation studies [18,19,20,21,22] have led to a significant change in PCa treatment. The Uro-Oncology Study Group of the Italian Association of Radiotherapy and Clinical Oncology (AIRO) has been committed for years to collect a large number of consecutive cases to allow the evaluation of the clinical impact of RT planning and delivery advance on PCa patients. This effort led to the publication of the previous “Pattern Of Practice” (POP) studies [5,6]. The present analysis addresses the 2004–2011 period; main features of this period were the increasing adoption of hypofractionated schedules and image guided radiation therapy.

While acknowledging the retrospective nature of our analysis and the different related biases, some interesting information can be extracted. RT is confirmed as the preferred choice in older patients, as demonstrated by the median age of patients included in the present study (72 years, range 46–88). Almost 60% of patients had low or intermediate risk PCa (D’Amico classification); notably, 1152 patients (46%) had a Gleason score 2 through 6, with a lower percentage of Gleason score 2–5 in POPIII compared the POP II series (Table 1), which is likely a consequence of one of the major changes between the two revisions of 2005 [23] and 2014 [24] of the grading consensus of the International Society of Urologic Pathology (ISUP); the 2014 revision recommended that Scores 2–5 should no longer be assessed on needle biopsies. This finding parallels the observed reduction in the number of patients with pre-RT PSA values in excess of 20 ng/mL and the 11% increase in those diagnosed with Gleason sum ≥7 in the POP III series, and points to a more accurate selection of patients.

More than 50% of the patients were staged using abdominal CT scan with i.v. contrast, bone scan, and endorectal ultrasound, in agreement with current guidelines [13,14,15,16], which do not recommend the use of these diagnostic procedures in low risk group patients while they can be performed in intermediate risk ones. The percentage observed is therefore reasonable in our population, mostly represented by intermediate risk (30%) and high-risk patients (40%), as some patients with intermediate risk cancer did not actually undergo these procedures. On the other hand, endorectal and/or abdomino-pelvic magnetic resonance imaging (MRI) was used in a very low number of patients. These findings would likely be different nowadays, in view of the national and international guidelines’ recommendation to implement the use of multiparametric MRI (mpMRI) before biopsy in daily routine [13,14,15,16].

Doses to the prostate were significantly higher in comparison with POP II (Figure 1). These data seem to be clearly influenced by dose escalation studies published since 2002 [18,19,20,21,22], which showed a clinical benefit for doses >70 Gy in terms of disease control. For the first time, a small number of patients (107, 5%) was treated using moderate hypofractionation, confirming the increasing role of this strategy, particularly during the second half of the study period; this trend was subsequently confirmed with the publication of relevant positive studies with long term results [25,26,27,28]. Ninety-five percent of patients were treated only to prostate ± seminal vesicles, thus confirming the trend towards the omission of upfront prophylactic pelvic irradiation already observed in the POP II study, although several patients with high-risk disease (about 40%) were included in both series. These data reflect the lack of clear evidence for a clinical benefit of prophylactic pelvic irradiation (PPI), even in the setting of high-risk patients. An ongoing multicenter study (PROEPI) promoted by the AIRO Uro-Oncology Study Group will likely result in additional insight in this scenario; given also that data from a randomized Phase III clinical trial favoring the use of PPI in high and very high risk patients have been recently published [29].

Almost a quarter of the patients were treated with IMRT, while only a smaller fraction of them (13%) was treated with IGRT. The photon beam energy mainly used was ≤10 MV (1479 cases, 59%); this finding confirmed the conclusion of the previous study that photon beams between 10–18 MV were progressively abandoned, paralleling the increasing use of IMRT.

Overall, the treatment was well tolerated, as the majority of patients experienced only mild to moderate toxicities. In comparison to POP II, patients in the current series showed better tolerance to RT treatment with a lower crude rate of acute toxicity ≥ grade 2. Reported grade 4 acute urinary toxicity remained as low in POP III as it was in POP II (Figure 3). When analyzing the crude maximum late toxicity rates, most patients experienced mild toxicity (grade 2), while only 7 patients experienced grade 4 toxicity (6 patients GU and 1 GI, respectively) (Figure 4). Unexpectedly, patients treated with a total dose (EqD2) ≤ 74 Gy developed a higher incidence of acute and late toxicity, the likely reasons being the larger CTV-PTV margins used in this cohort in absence of IGRT.

Toxicity outcomes in our population are comparable with those reported in the contemporary literature. The review published by Ohri et al. [30] on nearly 12,000 PCa patients treated with different RT doses (median dose 72 Gy) and techniques found a median rate of late toxicity ≥ G2 of 15% and 17% for GI and GU events, respectively. On the other hand, median rate of late toxicity ≥ G3 was only 2% and 3%, respectively. In accordance with a plethora of previous data [19,21,31,32,33], also in this comparison of consecutive series, an increase in the prescription dose seems to be slightly but consistently associated with an increased toxicity.

In our series, no substantial reduction in the incidence of acute and late toxicity was documented in patients treated with IMRT, probably due to the relatively small number of patients treated with this technique and to the absence of a simultaneous homogenous reduction of the CTV-PTV margins while increasing the dose among them.

This issue is still highly debated in the scientific community, so that in the NCCN guidelines [13] the real impact of IMRT does not seem to be completely elucidated, while the use of IGRT is recommended. From the available literature, IMRT seems to reduce the rate of gastrointestinal toxicity, as confirmed in several studies [34,35,36]. More recently Barelkowsky et al. [37] reported acute G2 and 3 GU toxicity in 39.8% and 1.1% of patients, respectively, acute G2 and 3 GI toxicity in 12.5% and 0%, late G2 and 3 GU toxicity in 19.3% and 4.5%, and late G2 and 3 GI toxicity in 4.5% and 1.1%. All patients were treated by IGRT with Tomotherapy in a radical setting and no toxicity >G3 was observed. Additionally Detti et al. [38] recently published a retrospective analysis on 394 patients treated with radical high-dose EBRT (mean total dose 79 Gy, standard fractionation) for localized PCa using IMRT with daily IGRT. The authors reported a low toxicity profile, with 51.8% G2 GU acute side effects and only 2 cases (0.5%) of G3 GU toxicity and no acute G3 GI toxicity. Late toxicity rate was very low, with only 1 patient (0.2%) who developed G3 GU toxicity. The authors concluded that IGRT should be mandatory for the treatment of PCa when high dose EBRT is used. Results of the randomized IGIM trial (www.igimtrial.unimore.it), which recently completed accrual, might further corroborate this recommendation. Considering the data of the three retrospective series (POP I, POP II, POP III), the cumulative incidence of late toxicity (at five years) reported for POP II [6] (24.8%) was lower than that reported for POPI [5] (31–37%), while in POP III there was an increase (30%). This trend could be explained by the fact that in the transition from POP I to POP II the dose prescription had not yet changed and remained almost similar among the two series, while conformational techniques were introduced in the daily practice. In POP III, on the other hand, a significant dose escalation with better outcomes in terms of BRFS was achieved without a simultaneous adjustment of the treatment techniques, especially in terms of a reduction in CTV-PTV margins, at least in the first half of the study period (Table 4).

## 5. Conclusions

This longitudinal comparison of prostate RT practice patterns reported for a national observational cohort provides a clear summary of the changes in recruitment, diagnosis, adopted treatment techniques, prevalent dose prescriptions, and toxicity profile in the different time periods. Changes in histopathological assessment and staging procedures that might have caused stage migration have also been observed. The data further confirm that an increasing use of conformal techniques with IMRT and IGRT allowed the increase of prescription doses consistently up to >70 Gy. PCa radiation therapy evolved simultaneously during the study period towards an increasing use of hypofractionation, reduced CTV-PTV margins, IMRT, and IGRT techniques, showing favorable results in terms of disease control and treatment tolerance.

## Figures and Tables

**Figure 1 cancers-13-02702-f001:**
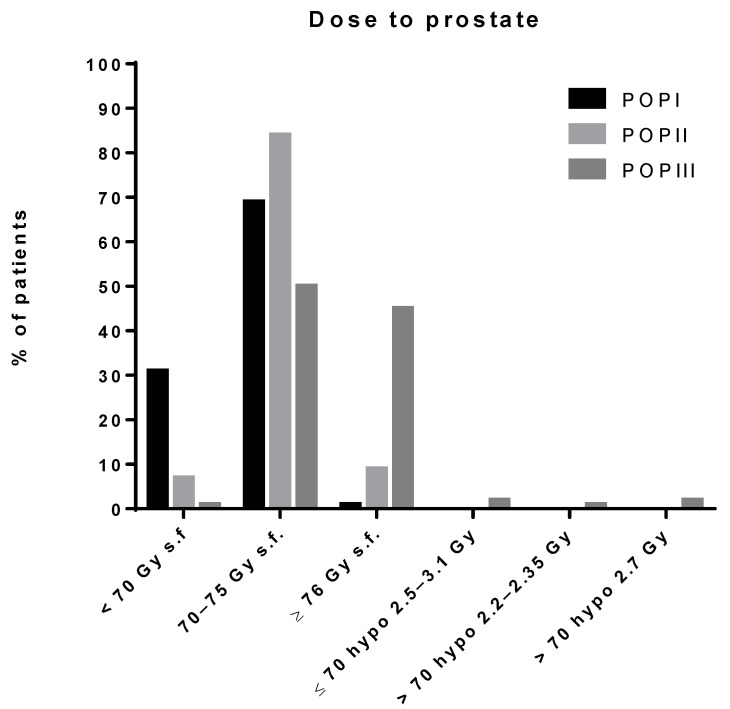
Dose delivered to prostate: POP I (1995–1998), POP II (1999–2003), POP III (2004–2011). Hypo: hypofractionation, s.f.: standard fractionation.

**Figure 2 cancers-13-02702-f002:**
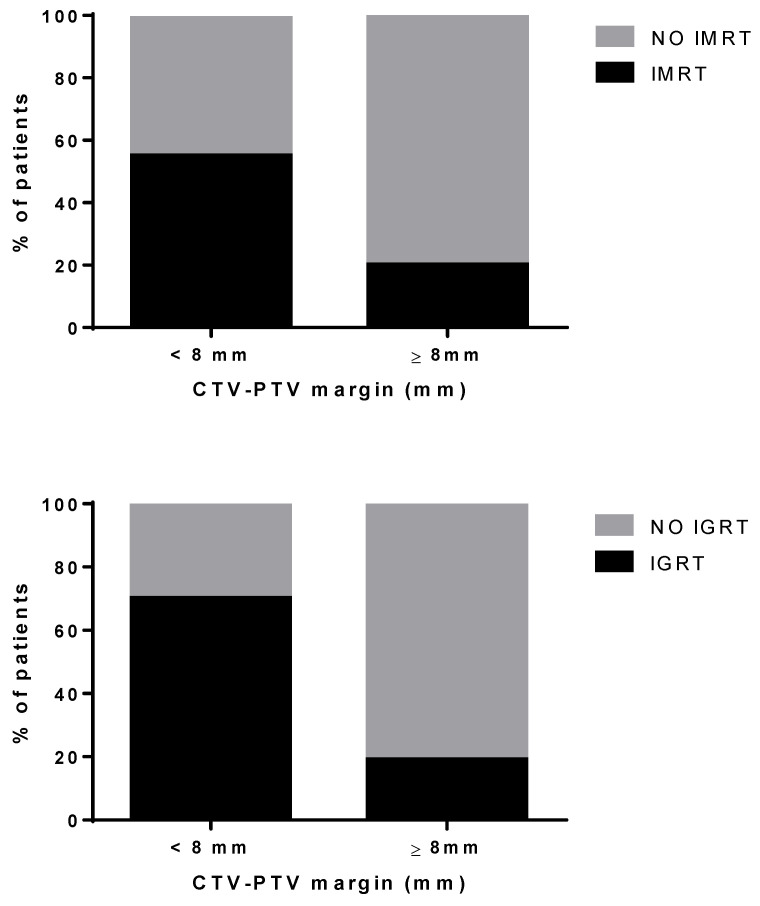
Percentage of treatment plans with CTV-PTV margin ≥ or <8 mm in IMRT vs. No IMRT treatments and IGRT vs. No IGRT treatments.

**Figure 3 cancers-13-02702-f003:**
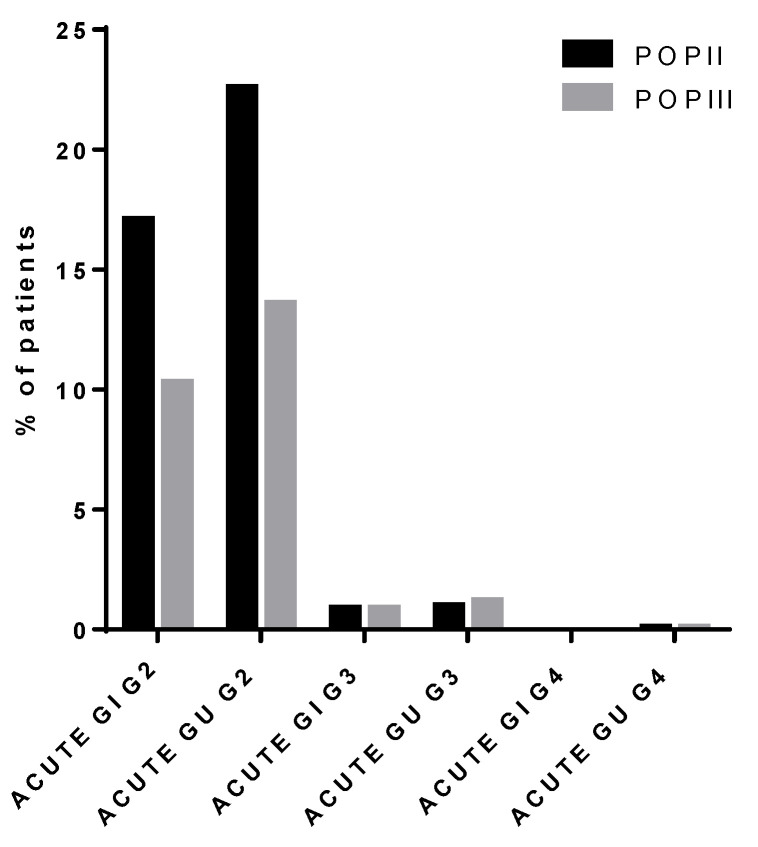
Acute toxicity, POP II (1999–2003), POP III (2004–2011).

**Figure 4 cancers-13-02702-f004:**
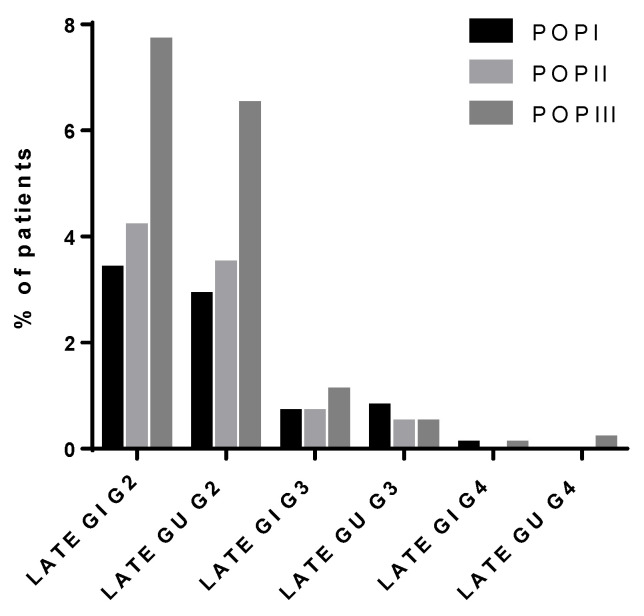
Crude rate of maximal late GU and GI toxicity through three series: POP I (1994–1998), POP II (1999–2003), POP III (2004–2011).

**Figure 5 cancers-13-02702-f005:**
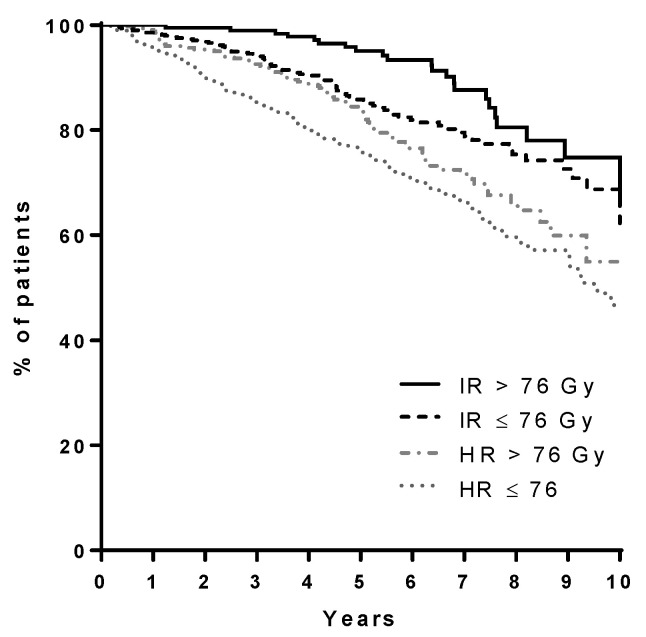
BRFS among intermediate risk patients (IR) and high risk patients (HR) treated with a total dose (EqD2) ≤ 76 Gy and > 76 Gy, log-rank *p* < 0.0001.

**Table 1 cancers-13-02702-t001:** Clinical features of the patients enrolled in the three subsequent surveys.

Clinical Feature	POPI 1995–1998 1005 Patients (No., %)	POPII 1999–2003 3001 Patients (No., %)	Current Series (POPIII) 2004–2011 2525 Patients (No., %)
Median age (range, yr)	70 (range, 28–89)	71 (range, 46–86)	72 (range, 46–88)
T stage			
1	99 (10)	597 (20)	866 (34)
2	577 (57)	1732 (58)	1177 (47)
3	284 (28)	621 (21)	450 (18)
4	17 (2)	29 (1)	11 (<1)
Unknown	28 (3)	5 (>1)	21 (<1)
N stage			
N+	32 (3)	75 (3)	42 (2)
N0	504 (50)	2502 (83)	1761 (70)
Nx	469 (47)	424 (14)	722 (28)
Grade group (ISUP)			
Group 1 (Gleason score ≤ 6)	Not evaluable	1380 (46)	1152 (46)
Group 2 (Gleason score 3 + 4 = 7)	Not evaluable	460 (16)	548 (22)
Group 3 (Gleason score 4 + 3 = 7)	Not evaluable	279 (9)	254 (10)
Group 4 (Gleason score 8)	Not evaluable	314 (10)	356 (14)
Group 5 (Gleason scores 9–10)	Not evaluable	153 (5)	188 (7)
Unknown	Not evaluable	409 (14)	27 (1)
Patients with Gleason score 2–5			
	Not evaluable	814 (27)	163 (6)
Gleason sum			
2–6	606 (61)	1676 (56)	1152 (46)
7	142 (14)	777 (26)	805 (32)
8–10	114 (11)	506 (17)	543 (22)
Unknown	143 (14)	42 (1)	25 (1)
Baseline PSA (ng/mL)			
<1	0 (0)	0 (0)	24 (<1)
1–10	345 (34)	1292 (43)	1481 (59)
10.1–20	277 (28)	886 (30)	617 (24)
>20	326 (32)	810 (27)	395 (15)
Unknown	57 (6)	13 (<1)	8 (<1)
Median baseline PSA (ng/mL)	13	12	8.2
D’Amico category risk			
Low risk	34 (3)	516 (17)	739 (29)
Intermediate risk	342 (33)	1057 (35)	756 (30)
High Risk	432 (43)	1356 (46)	1016 (40)
Not assessable	197 (21)	72 (2)	14 (<1)

**Table 2 cancers-13-02702-t002:** Diagnostic procedures adopted in the three subsequent surveys.

Diagnostic Procedures	POPI 1995–1998 1005 Patients (No., %)	POPII 1999–2003 3001 Patients (No., %)	Current Series (POPIII) 2004–2011 2525 Patients (No., %)	Chi-Square Test 1999–2003 Serie vs. 2004–2011 Serie
Abdominal/pelvic computed tomography				
No	395 (39)	798 (27)	1186 (47)	*p* < 0.001
Yes	610 (61)	2203 (73)	1339 (53)	
Abdominal/pelvic magnetic resonance				
No	867 (86)	2808 (94)	2335 (92)	*p* = 0.171
Yes	138 (14)	193 (6)	190 (8)	
Transrectal ultrasound				
No	254 (25)	385 (13)	809 (32)	*p* < 0.001
Yes	751 (75)	2616 (87)	1716 (68)	
Magnetic resonance with endorectal coil				
No	911 (91)	2714 (90)	2251 (89)	*p* = 0.114
Yes	94 (9)	287 (10)	274 (11)	
Bone scan				
No	86 (9)	471 (16)	1181 (47)	*p* < 0.001
Yes	919 (91)	2530 (84)	1344 (53)	
Staging lymphadenectomy				
No	928 (92)	2945 (98)	Not evaluated	\\
Yes	77 (8)	56 (2)	Not evaluated	
Choline PET				
No	Not evaluated	Not evaluated	2476 (98)	\\
Yes	Not evaluated	Not evaluated	49 (2)	
TURP				
No	Not evaluated	2741 (91)	2299 (91)	\\
Yes	Not evaluated	259 (9)	215 (8)	
Unknown	Not evaluated	1 (<1)	11 (<1)	

**Table 3 cancers-13-02702-t003:** Therapeutic procedures adopted in the three subsequent surveys.

Therapeutic Feature	POPI 1995–1998 1005 Patients (No., %)	POPII 1999–2003 3001 Patients (No., %)	Current Series (POPIII) 2004–2011 2525 Patients (No., %)
Dose to prostate			
<70 Gy s.f	315 (31)	218 (7)	20 (<1)
70–75 Gy s.f.	689 (69)	2531 (84)	1257 (50)
≥76 Gy s.f.	1 (<1)	252 (9)	1136 (45)
≤70 hypo 2.5–3.1 Gy Dmean to prostate EqD2 (α/β = 1.5): 78 Gy *	Not performed	Not performed	47 (2)
>70 hypo 2.2–2.35 Gy Dmean to prostate EqD2 (α/β = 1.5): 79 Gy *	Not performed	Not performed	20 (<1)
>70 hypo 2.7 Gy Dmean to prostate EqD2 (α/β = 1.5): 84.2 Gy *	Not performed	Not performed	40 (2)
Unknown			5 (<1)
Dmean to prostate (Gy) **			
Conventional fractionation	69	71.6	74.1
EqD2 (α/β = 1.5)			74.5
Treated volume			
No pelvic irradiation	784 (78)	2497 (83)	2416 (95)
Pelvic irradiation	221 (22)	504 (17)	107 (4)
Unknown			2 (<1)
Beam energy, MV			
≤10	62 (6)	174 (6)	1479 (59)
10–18	421 (42)	2026 (68)	613 (24)
≥18	522 (52)	252 (8)	283 (11)
Unknown	-	549 (18)	150 (6)
Conformal techniques			
2D-RT	593 (59)	725 (24)	-
Static 3D-CRT	412 (41)	2174 (72)	1953 (77)
Dynamic 3D-CRT	Not performed	95 (3)	4 (<1)
IMRT	Not performed	7 (<1)	568 (22)
IGRT			
No	All	All	2188 (87)
Yes	None	None	337 (13)
CTV-PTV margin (mm)			
<8			625 (25)
≥8–10			1747 (70)
Unknown	All	All	153 (5)
Concomitant hormone therapy			
No	201 (20)	644 (22)	581 (23)
Yes	804 (80)	2226 (74)	1911 (76)
Unknown	-	131 (4)	33 (1)

sf = standard fraction (1.8–2 Gy); hypo = hypofractionated regimens; * D mean = mean prostate dose in the subgroup; ** All patients.

**Table 4 cancers-13-02702-t004:** Techniques, doses, CTV-PTV margins and toxicity in the first and second half of the accrual period; (*) = data unknown for 153 cases. Pts = patients.

Treatment Features and Toxicity	Period		*p*
	2004–2007 (1246 Patients, 49%)	2008–2011 (1279 Patients, 51%)	
Pts treated with IMRT (%)	7%	38%	*p* < 0.001
Pts treated with IGRT (%)	7%	19%	*p* < 0.001
Pts treated with EqD2 ≥ 78 Gy (%)	22%	28%	*p* < 0.001
Mean dose (EqD2) (±SD) Gy	74 Gy (± 3.6)	75 Gy (± 3.5)	*p* < 0.001
CTV-PTV margin < 8 mm (%) ^(*)^	15%	38%	*p* < 0.001
Hypofractionation (%)	1%	7%	*p* < 0.001
Acute GI (≥G2) toxicity (%)	15%	8%	*p* < 0.001
Acute GU (≥G2) toxicity (%)	16%	14%	*p* = NS
Late GI (≥G2) toxicity (%)	13%	13%	*p* = NS
Late GU (≥G2) toxicity (%)	14%	8%	*p* < 0.001

## Data Availability

The POPIII participating data are available only to the collaborating scientists within the study and are stored in shared file (excel) after anonimization. The data presented in this study are available on request from the corresponding author.

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
