# Peer review of "How Has Prostate Cancer Radiotherapy Changed in Italy between 2004 and 2011? An Analysis of the National Patterns-Of-Practice (POP) Database by the Uro-Oncology Study Group of the Italian Society of Radiotherapy and Clinical Oncology (AIRO)"

_cancers, 2021, doi:10.3390/cancers13112702_

Round 1
Reviewer 1 Report
This is a report which presents data on the prostate cancer radiotherapy. Specifically, authors present the results of the years between 2004 to 2011, called POPIII and compare these with previous previuous years, i.e. 1980-2003 divided in two studies called POPI and POPII.
The objective of the report is to show the outcome of the radiotherapy on patients affected with prostate cancer, concerning characteristics of the therapy, such as dose, intensity etc. Outcomes include the survival in terms of overall, relapse free and disease specific survival of patients according to Kaplan-Meier method. This method was also used to present the side effects and toxicity that affected the patients after treatment.
The report is very well presented and results are valuable for the treatment of prostate cancer and with high scientific soundness.
Author Response
Dear Reviewer,
We are grateful to you for your time and constructive comments.
As you suggested, we applied some corrections to obtain a more fluent English language.
Furthermore I corrected some editorial refusal in Table 1 (Page 3), 2 (Page 4) and Figure 2 (Page 7).
Changes in the manuscript are highlighted using track changes method.
Yours sincerely,
On behalf of the co-authors
Alessio Bruni
Reviewer 2 Report
This third series of the Patterns of Practice (POP III) study includes the prostate cancer (PCa) patients treated with EBRT in Italy during the time period 2004-2011. The clinical outcomes of POP-III has been compared to POP-II and found to be comparable. Dose escalation improved the tumor control at the cost of increased toxicity (Garde 2-3 GI and GU). Old series with less than 74Gy exhibited higher acute and late toxicities which was attributed to lack of IGRT. CTV to PTV margin greater than 8mm is also related to increased toxicity. The results and conclusions of POP-III are similar to other major studies found in the literature. Overall, this paper is well written and may be considered a good supportive report to other published studies in PCa treatment with EBRT.
Author Response
Dear Reviewer,
On beahlf of all the authors I am grateful to you for your time and constructive comments.
As requested, we applied some corrections to obtain a more fluent English language.
Furthermore I corrected some editorial refusal in Table 1 (Page 3), 2 (Page 4) and Figure 2 (Page 7).
Changes in the manuscript are highlighted using track changes method.
Yours sincerely,
On behalf of the co-authors
Alessio Bruni